# Furanic Humins from Biorefinery as Biobased Binder for Bitumen

**DOI:** 10.3390/polym14051019

**Published:** 2022-03-03

**Authors:** Anna Sangregorio, Nathanael Guigo, Luc Vincent, Ed de Jong, Nicolas Sbirrazzuoli

**Affiliations:** 1Institut de Chimie de Nice, Université Côte d’Azur, CNRS, UMR 7272, 06108 Nice, France; sangregorioanna@gmail.com (A.S.); luc.vincent@univ-cotedazur.fr (L.V.); nicolas.sbirrazzuoli@univ-cotedazur.fr (N.S.); 2Avantium N.V., Zekeringstraat 29, 1014 BV Amsterdam, The Netherlands; ed.dejong@avantium.com

**Keywords:** humins, side-products, bitumen, rheology, biorefinery

## Abstract

To decrease the environmental impact of bitumen, more sustainable binders should be proposed. This study emphasizes how industrial humins co-produced during the biorefining of carbohydrates can be employed as a macromolecular binder for bitumen. Humins are heterogeneous polyfuranic compounds, and they were mixed at 50 wt% with bitumen. When the non-water-soluble fractions of humins were employed (Hns), no variation of the chemical structure was observed in FTIR spectra after the mixing. The DSC investigations showed that the crystallization of aromatic fractions in bitumen shifted to higher temperature for humins’ modified bitumen. The thermogravimetric data highlighted that the presence of humins or Hns in bitumen can lead to mass loss below 200 °C. The rheological investigations highlighted some key advantages of using humins or Hns with bitumen. At high temperatures, the storage modulus of the modified bitumen is increased and shows lower susceptibility to variations in frequency. At low temperatures, the phase angle of Hns-modified bitumen is lower than that of bitumen, suggesting less temperature susceptibility as a consequence of a cross-linked network formation.

## 1. Introduction

Nowadays, paving-grade bitumen is almost exclusively obtained as the vacuum residue of petroleum distillation. About 95% of the bitumen produced is employed as a binder for mineral aggregates to form asphalt mixes used in the paving industry. The most critical properties for obtaining suitable bitumen are mainly rheological. Bitumen viscosity at high temperature (around 160 °C) should be low enough to easily mix it with the aggregates. Bitumen should also be stiff enough at the highest pavement temperatures (around 60 °C) to resist rutting and soft enough at the lowest temperatures (around −20 °C) to avoid cracking. However, obtaining these properties is quite complicated, and thus the use of additives and modifiers is more and more common [1]. Bitumen structure and chemistry is very complicated as many different chemicals are present. Conventionally, bitumen can be considered as being composed of four different fractions, called saturates, asphaltenes, resins, and aromatics (identified as the “SARA” fractions). Saturates are mainly aliphatic, with a H/C ratio close to 2. Saturates are characterized by a very low glass transition temperature, *T*_g_, around −70 °C, and few crystalline linear n-alkanes. Aromatics, together with resins, are the most abundant constituent of bitumen. Aromatics have a *T*_g_ around −20 °C and mainly determine the final *T*_g_ of the parent bitumen. Aromatics are mainly constituted by non-polar carbon chains in which the unsaturated ring systems dominate. Resins have a similar composition as asphaltenes but with lower molecular weight and less complex aromatic structure. Resins are very polar components, acting as stabilizers for asphaltenes. Asphaltenes are highly polar, complex aromatic macromolecules with a tendency to interact and associate, mainly determining the surface activity and the adhesion of bitumen onto mineral aggregates. They do not display any thermal transitions up to 200 °C [2,3]. As mentioned above, bitumen is mainly derived from petroleum. However, the depletion of petroleum resources and increasing construction costs and strong demand for asphalt are encouraging the development of binders from renewable resources [4,5]. In addition, bitumen causes a severe negative impact on the environment and on the health of pavement workers as, during the paving process, vapors containing greenhouse gases and toxic volatiles are released [6]. Thus, the need to develop a more sustainable alternative to decrease the use of bitumen is becoming increasingly urgent. Three different categories of binders can be distinguished: (i) direct alternative binder, corresponding to 100% replacement; (ii) bitumen extender, corresponding with 25–75% asphalt replacement; (iii) bitumen modifier, corresponding with less than 10% replacement. Among the most promising binders to decrease the consumption of bitumen are bio-oils, vegetable oil, and waste materials. Bio-oils can be extracted from organic plants and residues. Waste materials such as waste cooking oil, waste engine oil, and biodiesel residues might also be a promising alternative [5,7,8]. Several oils have already been tested as a type of modifier to improve the low temperature performance of bitumen, decrease its cost, and reduce the risk of environmental pollution. Waste cooking oil has been proposed to decrease the percentage of petroleum-based bitumen and to counter the generation of significant amounts of waste oil that might generate problems when not disposed of properly. Results show an improvement in low-temperature performance [9,10]. Other studies focused on the use of vegetable oils as a binder [11]. Bitumen binder containing up to 30% castor oil was also studied, showing a variety of positive effects with different properties [12]. Systematic studies on opportunities for the use of biobinders using different bio-oils derived from pyrolysis of different lignocellulosic biomasses was undertaken [13]. Generally, these types of bio-binder can improve the high-temperature performance of matrix asphalt, whereas the low-temperature performance of matrix asphalt would be slightly reduced [13,14]. Isolated lignin is a polyphenolic macromolecule that is a waste-stream product from the pulp and paper industry. Its combination with bitumen has been extensively explored in recent decades. When lignin is combined with bitumen, it allows better rutting performance at high temperature and better aging resistance of asphalt mixtures [15]. A recent life cycle assessment showed that the impact on climate change due lignin-based asphalts could be 30–75% lower than the impact due to conventional asphalts [16].

In biorefinery operations of plant biomass, the conversion of carbohydrates and especially abundant monosaccharides (e.g., glucose and fructose) into key furanic building blocks (furfural, hydroxymethyl furfural and its alkyl-ethers) is accompanied by the formation of a tarry black viscous co-product known as humins [17]. The management of humins is therefore of particular importance for the value chain of furandicarboxylic acid (FDCA), which aims at developing and producing promising polyesters such as PEF at industrial scale. Industrial humins as heterogenous macromolecular compounds can be considered *furanic lignins* and therefore should find sustainable pathways of valorization. This is already the case for so-called non-furanic humins (composed of humic and fulvic substances in soil) that can be processed with tannins to prepare non-isiocyanate wood adhesives [18]. Recent examples have demonstrated that furanic humins can be employed in substantial amounts to prepare durable wood panels [19], biobased composites [20], or foams [21,22] or as a UV shielding agent for thin PVA films [23]. It should be noted that humins can be considered reactive thermoset resins pending appropriate activation upon heating [24]. The similarity between humins and lignins (i.e., abundant biobased macromolecular wastes) thus suggests that humins from biorefineries could be another sustainable candidate in the scenario of finding bioderived binders for bitumen. To our knowledge, no studies have been devoted to this valorization strategy except a recent patent application from Avantium [25].

In this study, humins were used as a bioderived binder to be mixed with bitumen in order to decrease its environmental impact. The aim is to understand if large quantities of humins could be mixed with bitumen, without compromising the final rheological properties but rather improving them. Possible interactions between the two components were studied by IR spectroscopy to achieve insights into the structures of the new biobased binder. Thermo-analytical techniques, such as DSC and TGA, were employed to obtain information on physical and chemical transition and degradation steps.

## 2. Materials and Methods

### 2.1. Materials

A commercial bitumen sample called Bitumen 40/60 was used. Bitumen Penetration Grade 40/60 is a standard penetration-grade bitumen usually used as a paving-grade bitumen suitable for road construction and for the production of asphalt pavements with superior properties. This grade of bitumen is mainly used in the manufacture of hot-mix asphalt for bases and wearing courses. The use 40/60 Bitumen allows functional properties to be achieved that you cannot be achieved achieve with a softer 70/100 bitumen.

Industrial humins, hereafter called “humins” were produced as a side-product of fructose conversion in the production process for FDCA by Avantium N.V. in their pilot plant (Geleen, The Netherlands).

Humins were heated in an oil bath and mixed at 80 °C for 20 min. Demineralized water, preheated at 80 °C, was added to humins at a ratio of humins/H_2_O of 50/50% in terms of weight and mixed under mechanical stirring. After 40 min, the mixture was taken out of the oil bath and allowed to cool down to room temperature. Then, two phases were observed: one solid residue and one supernatant liquid phase. The liquid phase (Hs) was poured and divided from the solid residue. The solid residue corresponding to insoluble phase of the crude humins are herein labeled as “Hns” (humins non-soluble). Hns was dried in an oven at 140 °C for 2 h to obtain a solid material and was ground into a powder.

Three different samples were prepared with the bitumen.

(i)Humins and bitumen were heated up at 80 °C for 20 min to decrease their viscosity. Afterward, the two materials were mixed at a ratio 50:50% in terms of weight and stirred for other 40 min at 80 °C. This sample is labeled as 50B/50Humins.(ii)Hns was mixed with bitumen and preheated for 20 min at 80 °C at a ratio of 50:50% wt. and stirred for other 40 min at 80 °C. This sample is labeled as 50B/50Hns.(iii)To exclude the effect of bitumen aging during the mixing process and thus evaluate accurately the modifier (Humins or Hns) effects, the base bitumen was treated under the same temperature conditions (i.e., at 80 °C for 60 min and mechanically stirred) as the modified ones and used for comparison.

### 2.2. Infrared Spectroscopy

A Bruker tensor 27–FTIR spectrometer equipped with a nitrogen-cooled MCT detector was used to characterize samples using a 1-reflection diamond ATR device. The spectrum of air was recorded as the background before each measurement. A total of 64 scans with a resolution of 2 cm^−1^ were recorded for each sample.

### 2.3. Differential Scanning Calorimetry (DSC)

Conventional differential scanning calorimetry measurements were performed on a heat flux Mettler-Toledo DSC-1, and STAR^©^ software was used for data analysis. Temperature, enthalpy, and tau lag calibrations were performed using indium and zinc standards. Five to ten milligrams of samples were placed in a 40 µL aluminum crucible and closed by a pan lid. The experiments were done under air flow (50 mL.min^−1^). Samples were measured using a regular DSC with a scanning temperature ranging from −80 °C to 140 °C. The heating rate employed was 30 °C·min^−1^.

### 2.4. Thermogravimetric Analysis (TGA)

The thermo-oxidative degradation of the samples was studied on a Mettler–Toledo TGA/SDTA 851e. Samples were measured at a heating rate of 10 °C min^−1^ under air flow (50 mL min^−1^). Mass calibration was performed using the masses of the internal calibration ring of the device. Temperature calibration was performed using indium and zinc standards.

### 2.5. Rheometry

The rheological properties of the samples were registered with a Thermo Scientific HAAKE MARS rheometer. Measurements were obtained on plate–plate geometry (25 mm diameter and 1 mm gap). Samples of the linear viscoelastic region were defined for each temperature by a strain sweep. Then, the material was characterized using a frequency sweep at a strain below the critical strain. The rheological responses were registered at decreasing angular frequency (10 to 0.1 Hz) for different temperatures (40 °C, 60 °C, 80 °C, 100 °C, 120 °C, 140 °C). Three different measurements per sample were recorded. Samples were also tested in temperature sweep mode, from −15 °C to 35 and from 80 °C to 180 °C, with a heating rate of 1 °C min^−1^.

## 3. Results and Discussion

### 3.1. FT-IR Analysis

#### 3.1.1. Mixture of Bitumen with Humins

Figure 1 shows the infrared spectra of unmodified bitumen and bitumen mixture with humins, and Table 1 highlights the major peak assignments. Appendix A shows a representative scheme of the main structures in the furanic humins investigated herein that were obtained from fructose conversion. A peak corresponding with -OH stretching at 3383 cm^−1^ is observed after modification. This corresponds to -OH groups in humins. Interestingly, this broad peak is slightly shifted towards higher wavenumbers when humins are mixed with bitumen, suggesting a modification of the H-bonding network. This shift to higher wavenumbers when dispersed in bitumen indicate that humins bear more free OH groups than in the crude humins. This might be the consequence of a dispersion of humins chain in bitumen media. The intense peaks between 2850 and 2920 cm^−1^ are typical of the -CH stretching of aliphatic chains in bitumen. No shifts are observed in this region after modification. The peak observed around 1605 cm^−1^ in unmodified bitumen is attributed to C=C stretching vibrations in aromatics. This peak appears as a shoulder in modified bitumen, as it overlaps the conjugated carbonyl groups containing humins, peaking around 1615 cm^−1^. This peak is slightly shifted toward lower wavenumbers after mixing with bitumen, suggesting a modification in the environment of these functional groups. Moreover, new peaks are formed in the 1700–1750 cm^−1^ region, related to modification of the C=O groups conjugated to alkene, which might indicate interactions between humins and bitumen or formation of new carbonyl groups. The -CH asymmetric deforming in CH_2_ and CH_3_, and -CH symmetric deforming in CH_3_ vibrations, were observed at 1458 cm^−1^ and 1375 cm^−1^, respectively. Both peaks can also be found after modification. The region between 1100 and 1200 cm^−1^ is usually associated with C-O stretching from ethers and esters. A quite significant shift of these peaks to higher wavenumbers is observed for humins after mixing with bitumen, suggesting a modification of the chemical environment of humins. These C-O bonds also behave similarly to the H-bond acceptor, and thus the shift can be correlated with the modification of the humins -OH and carbonyl groups.

#### 3.1.2. Mixture of Bitumen with Hns

Figure 2 shows the infrared spectra of unmodified bitumen, Hns, and bitumen mixture with Hns. All the peaks observed in Hns-modified bitumen that are not observed in unmodified bitumen can be attributed to Hns. Moreover, two peaks from bitumen are also evident in Hns-modified bitumen, corresponding to 1378 and 1455 cm^−1^. The shift to higher wavenumbers of C=O (carbonyls) and C-O (ether, esters), respectively, in the 1750 cm^−1^ and 1100–1200 cm^−1^ regions are not observed when Hns is mixed with bitumen. This would indicate that the water-soluble chains in humins (that are removed in Hns) are modified by mixing with bitumen. These modification observed in humins are most likely attributed to residual cross-linking when processed at relatively high temperatures. On the other hand, Hns is relatively stable during the mixing phase, since no particular shifts are observed after mixing with bitumen.

### 3.2. Thermal Behavior

Thermal properties were first studied using DSC. Appendix A shows the DSC scans of both crude humins, the soluble part of humins (Hs), and Hns. Appendix A shows the *T*_g_ values obtained from the evaluation of these DSC scans. Hns logically exhibits higher *T*_g_ compared to crude humins and Hs. Indeed, Hs gathers the shortest humins’ chains with polar groups whose *T*_g_ is considered lower than the more insoluble branched chains contained in Hns. Figure 3 shows the results for unmodified bitumen, 50B/50Humins, and 50B/50Hns. Unmodified bitumen shows a sigmoidal decrease at low temperature, with an inflection point at around −30 °C. This transition is mainly linked to the *T*_g_ of the aromatic fraction of bitumen and is spread over a wide range of temperature, from −50 °C to around 5 °C (Figure 3). However, this phenomenon overlaps with crystallization of the waxes fraction, which begins around 5 °C up to 20 °C. This transition occurs right after the *T*_g_ as higher mobility is necessary to allow crystallization. In both modified bitumen samples, the *T*_g_ seems to slightly increase, suggesting that Hns and humins, which have higher *T*_g_ than bitumen (Appendix A), partially mixed with bitumen, increasing the overall *T*_g_.

Figure 4 shows the thermogravimetric curves of unmodified and modified bitumen. The 50B/50humins sample shows a first mass loss of about 15% from 120 °C to 220 °C, corresponding with the loss of volatiles and water from condensation reactions in humins. In 50B/50Hns, the initial step of mass loss represents ~10 % and is slightly shifted to higher temperatures (i.e., from 140 °C to 260 °C). These residual condensations in the humins/bitumen mixtures are highlighted in Figure 4. The value of *T*_10%_ for modified bitumen is significantly decreased, in comparison with crude bitumen, which does not show a mass loss until around 300 °C. For unmodified bitumen, a first degradation step corresponding to ~15% mass loss is observed between 25 and 380 °C, with the maximum decomposition rate observed around 370 °C. This step is attributed to low-temperature oxidation (LTO) [26]. The same step is also observed in modified bitumen at lower temperatures. In this temperature range, lightweight hydrocarbons are oxidized and then are evolving. Humins and Hns might interfere with these chemical reactions, inducing faster oxidation. A second region is identified between 380 and 470 °C in unmodified bitumen, related with fuel decomposition (FD). The mass loss of about 37% in this temperature range is attributed to a combination of hydrocarbon combustion, which competes with thermal cracking reactions. Carbon-rich residue (coke) is formed at this stage [26]. For 50B/50Humins, the mass loss (~24%) attributed to FD is logically lower and occurs between 350 °C and 457 °C, which corresponds to a shift to lower temperature compared to bitumen. The mass loss attributed to this step is ~32% for 50B/50Hns-modified bitumen. Finally, bitumen shows a very fast degradation step of ~45% between 470 °C and 560 °C, identified as high-temperature oxidation (HTO). This region corresponds to the combustion of the remaining hydrocarbons and carbon residues [27]. It is interesting to note that this final step of mass loss occurs in a larger temperature range for the 50B/50Hns and 50B/50humins, suggesting that this decrease in the rate of combustion could be attributed to the aromatic rich residues in humins that are more difficult to oxidize compared to raw bitumen.

### 3.3. Rheological Behavior

#### 3.3.1. Frequency Scan

The frequency dependency of the complex viscosity at 60 °C is shown in Figure 5a. The three samples show an almost Newtonian behavior over the entire span of frequency studied, as the viscosity is almost independent of the frequency (Figure 5a). However, 50B/50Hns sample shows slightly higher values of viscosity compared to raw bitumen. On the other hand, 50B/50Humins sample displays a slightly lower viscosity. This might be explained by the fact that, at this temperature, humins viscosity is lower compared to Hns viscosity. Moreover, the presence of more polar and shorter chains in humins might decrease the number of interactions between macromolecules and thus decrease the viscosity. Figure 5b shows the variation in storage modulus with frequency. Frequency susceptibility is reduced in the case of 50B/50Humins, as the tendency of the curve is less steep, and its elastic response is also increased for low frequencies. For 50B/50Humins, the modulus is higher at low frequencies and lower at high frequencies. The 50B/50Hns sample shows very similar trends to unmodified bitumen, showing slightly higher storage modulus. Different behaviors are observed when investigating rheological properties at 140 °C.

As shown in Figure 6, modified bitumen shows a shear thinning behavior at low frequencies, while unmodified bitumen shows a Newtonian behavior, followed by a shear thickening behavior similar to unmodified bitumen. At a low frequency, the higher viscosity is related to increased humins and Hns interactions with bitumen, while increasing the frequency leads to the effect of entanglements dominating and viscosity decreasing [28]. All samples show a very non-uniform viscosity reaching high values for frequencies above 10 Hz. This is related to a significant increase of storage modulus for these values of frequency. Figure 6b shows the variation of storage modulus with frequencies. At low frequencies, between 0.1 and 10 Hz, modified humins show lower susceptibility to frequency variation as the value of storage modulus is almost constant. Moreover, the values of storage modulus for 50B/50Humins and 50B/50Hns are also higher compared with unmodified bitumen, suggesting that a new network was formed between bitumen and humins or Hns. At higher frequencies, above 10 Hz, the storage modulus significantly increases. This behavior might be linked with the formation or enhancement of transient networks of bitumen components, linked by hydrogen bonding and dipole–dipole interactions [28]. These results reveal a very similar behavior for humins and Hns at these specific temperatures.

#### 3.3.2. Temperature Scan

One of the most important parameters used for bitumen modification is to improve its rheological properties to avoid cracking caused by thermal shrinkage. This is a major concern for asphalt pavement in cold climate regions, as it can lead to major pavement damage. Improving the temperature performance of asphalt binders is an important focus [10]. Complex viscosity is shown in Figure 7as a function of temperature. In the low-temperature range, i.e., between −20 and 35 °C, the three samples show quite similar behaviors and viscosities. A Newtonian plateau is observed at low temperatures while, with increasing temperature, the viscosity decreases rather showing a shear thinning behavior. Figure 7a shows a higher viscosity with Hns modification and a lower decrease in viscosity with humins modification. Moreover, 50B/50Hns seems less susceptible to temperature changes, as viscosity is constant up to around 15 °C and then slowly starts decreasing. On the contrary, 50B/50Humins shows a higher susceptibility to viscosity as its value starts dropping from around −5 °C. These results might indicate that Hns interacts with bitumen, creating a stronger network, while humins act more as a plasticizer, decreasing interactions between chains and promoting the flowing of the chains. A different behavior is observed at high temperatures (Figure 7b). In the temperature range between 80 and 180 °C, both humins and Hns modification induce an increase in viscosity, indicating a stiffening effect. Moreover, the flatter slope for 50B/50Hns starting from 120 °C shows lower temperature susceptibility at higher temperature compared to unmodified bitumen. Both modified bitumens also show a less regular dependency on temperature, as both humins and Hns undergo a loss of volatiles and condensation reactions starting from 120 °C, which might induce abrupt changes in the viscosity value. The ideal asphalt mixture mixing and compaction viscosities, corresponding, respectively, with 1 and 10 Pa s, have been added in Figure 7b (green horizontal lines). The temperatures associated with these values are higher after modification with humins or Hns, as observed in other polymer-modified bitumens [4]. With dynamic shear rheology, it is possible to determine the complex shear modulus *G** and the phase angle *δ*. The *G** is a measure of the total resistance of a material to deformation when repeatedly sheared. *δ* is a measure of the viscoelastic character of the material. If *δ* equals 90°, then the binder can be considered purely viscous in nature, and conversely, a *δ* of 0° would represent an ideal elastic solid. Thus, it is an indicator of the relative amount of recoverable and non-recoverable deformation. The phase angle is generally considered to be more sensitive to the chemical structure and, therefore, the modification of bitumen than a complex modulus [29,30]. At low temperatures, polymer modification should lower creep stiffness of the bitumen, thus improving the resistance to thermal cracking. At high temperatures, additives should increase the stiffness and elasticity of the bitumen as a result of an increased storage modulus and a decreased phase angle. Indeed, the increase in the storage modulus and the decrease in the phase angle improve the rutting resistance of the pavement. Figure 8a shows the variation of *G** and *δ* with temperature between −15 and 35 °C. Sample 50B/50Hns shows less temperature susceptibility as observed from the less steep *G** curves both at low and high temperature, and the higher values of *G**. This is usually an indication of the formation of a polymeric network, which might be linked with cross-linking or entanglements.

At a low temperatures, humins and bitumen might physically interact, while at higher temperatures, chemical reactions might occur between Hns chains or between Hns chains and bitumen. Indeed, at high temperatures, variation of *G** with temperature appears to be more complex compared with unmodified bitumen. This might be linked with cross-linking reactions. As shown in Figure 8a, *δ* is shifted towards lower values for 50B/50Hns in the temperature range between −15 and 35 °C, indicating an increase in elasticity compared with unmodified bitumen. The same applies in the higher temperature span between 80 °C and 180 °C (Figure 8b) when the *δ* values in 50B/50Hns are lower than *δ* values of unmodified bitumen up to 160 °C. Figure 8b shows that the *δ* values in 50B/50Hns reach a maximum and a minimum. This is usually observed when sufficient polymer networks are formed in the modified bitumen [30,31]. This is a positive effect, as it indicates improvement in elasticity, which would result in improvement in resistance to deformation of the asphalt. For the 50B/50Humins sample, higher values of *δ* were observed at low temperatures, while a decrease in *δ* was observed at higher temperatures, again up to 160 °C. *G** is lower compared with unmodified bitumen at low temperatures and higher at high temperatures. The presence of a more polar fraction in humins might indeed have negative effect on humins-bitumen network at low temperature, while cross-linking and loss of volatiles at high temperature might contribute to an increase in *G**.

## 4. Conclusions

This study investigated the possibility of using crude humins or Hns (i.e., the most apolar humins fractions obtained after separation) as a biobased binder to decrease the large consumption of bitumen. Partially substituting bitumen with humins would have a positive effect on the environmental footprint of the binder and could be a new valorization solution for humins. FT-IR data show that the chemical environment of C=O and C-O bands in the humins’ structure changes after mixing with bitumen, most likely due to thermal treatment at 160 °C. Indeed Hns—which contains much less reactive counterparts susceptible to further cross-linking at higher temperature—does not highlight major variations after mixing with bitumen. The *T*_g_ of bitumen is shifted to higher temperatures when mixed with humins or Hns, suggesting a partial interconnection between the two networks. In the same line, TGA scans reveal that humins and Hns accelerate the LTO of bitumen below 370 °C. On the other hand, the aromatic carbons present in humins residues at high temperature (>470 °C) slow down the HTO of bitumen. The mixing of humins or Hns with bitumen induces modification in the rheological behavior. At 60 °C, both 50B/50Humins and 50B/50Hns remain in a Newtonian regime. However, a shear thinning behavior is observed at 140 °C in the presence of Hns or humins, especially in the lower frequency range (<10 Hz). The lower viscosity susceptibility for 50B/Hns compared to 50B/50humins at a low temperature (<35 °C) suggests that Hns creates a stronger network with bitumen, while the compounds with the lowest molecular weight in humins might instead plasticize the bitumen. At high temperatures (>80 °C), both 50B/50Hns and 50B/50Humins show higher values of complex modulus and viscosity compared to unmodified bitumen. These observations at low and high temperatures can be directly connected with practical applications. The utilization of humins might increase the softness at low temperatures, thus preventing cracking. Moreover, Hns would also have a positive effect, as it would decrease the temperature susceptibility of bitumen at very low temperatures (<35 °C). Modified bitumens show an increase in the storage modulus and a decrease in the phase angle at high temperatures, which should improve the rutting resistance of the pavement, supporting the integration of humins or Hns. However, both materials increase the viscosity at high temperature, thus implying the need to increase the mixing temperature when processing bitumen.

## Figures and Tables

**Figure 1 polymers-14-01019-f001:**
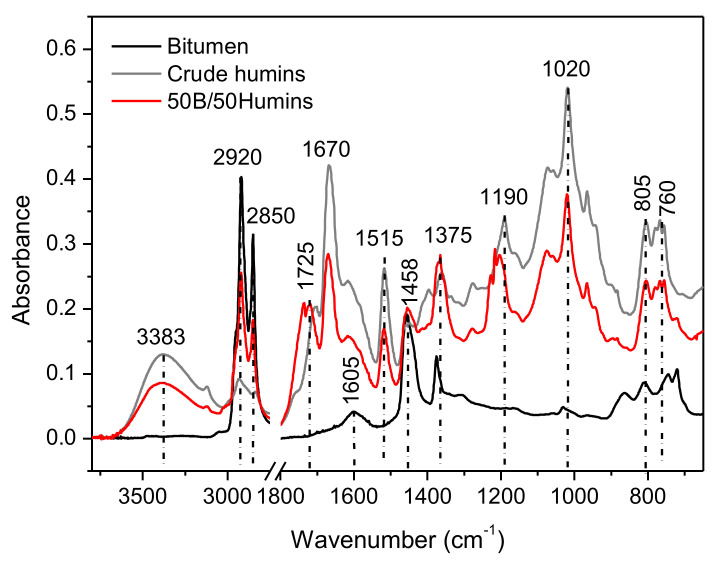
FTIR spectra of bitumen (black line), crude humins (grey line), and mixture bitumen/humins 50/50 (red line).

**Figure 2 polymers-14-01019-f002:**
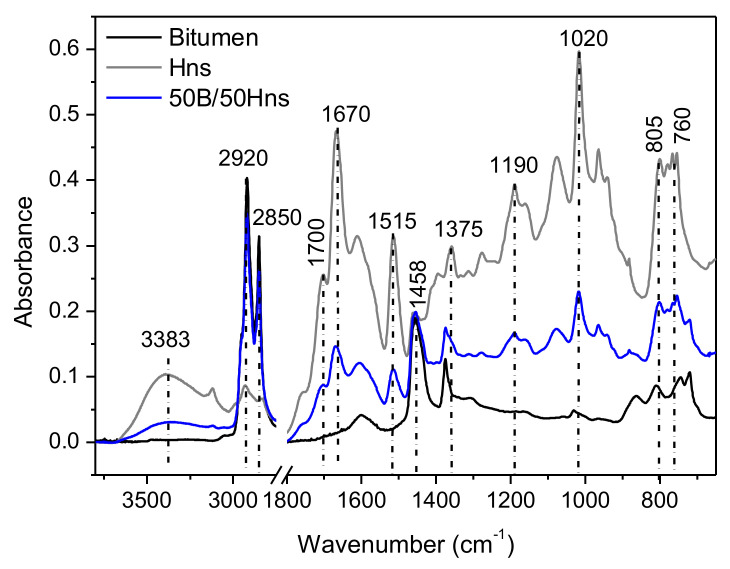
FTIR spectra of bitumen (black line), Hns (grey line), and mixture bitumen/Hns 50/50 (blue line).

**Figure 3 polymers-14-01019-f003:**
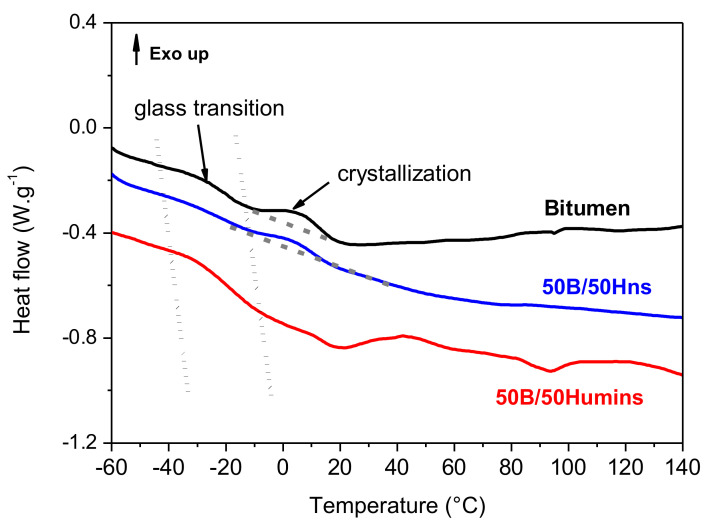
DSC scans of bitumen (black line), 50B/50Humins (red lines), and 50B/50Hns (blue lines). The dotted lines represent the temperature range for the glass transition. An estimation of the baseline is indicated by dash grey line to highlight the crystallization peak.

**Figure 4 polymers-14-01019-f004:**
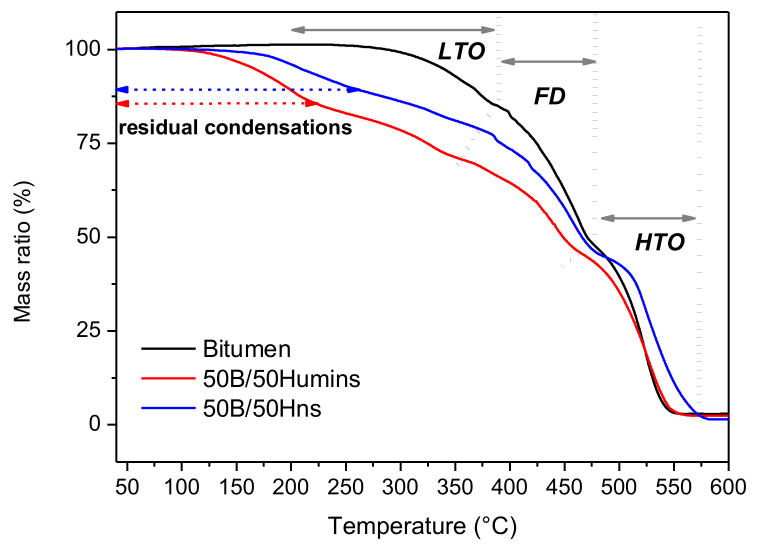
TG curves of bitumen (black line), 50B/50Humins (red line), and 50B/50Hns (blue line). The dotted lines separate the different regions of decomposition.

**Figure 5 polymers-14-01019-f005:**
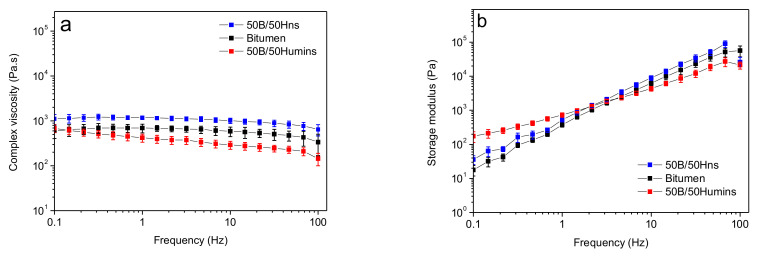
(**a**) Variation in the complex viscosity as a function of angular frequency at 60 °C; (**b**) variation in the storage modulus as a function of angular frequency at 60 °C. Error bars indicate the standard deviation on the means of 3 experiments.

**Figure 6 polymers-14-01019-f006:**
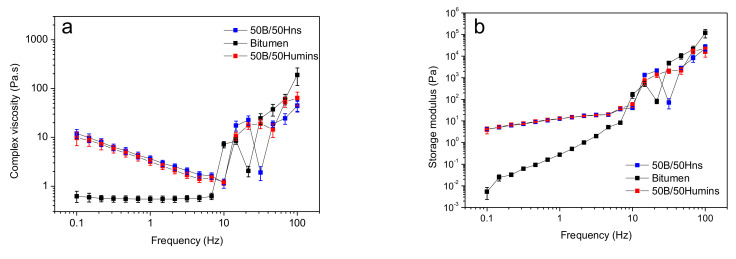
Variation of the complex viscosity (**a**) and storage modulus (**b**) as a function of angular frequency at 140 °C. Error bars indicate the standard deviation on the means of 3 experiments.

**Figure 7 polymers-14-01019-f007:**
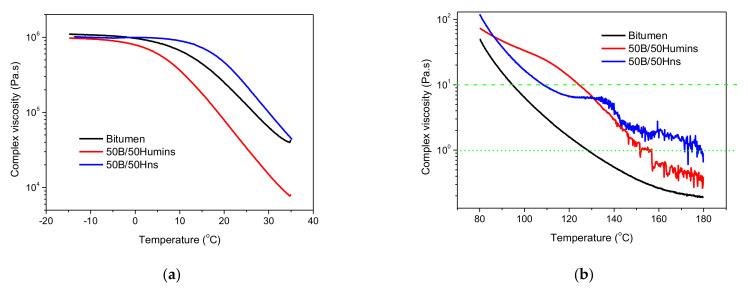
(**a**) Variation of the complex viscosity between −15 and 35 °C and (**b**) between 80 to 180 °C for crude humins (black line) 50B/50Humins (red line) and 50B/50Hns (blue line).

**Figure 8 polymers-14-01019-f008:**
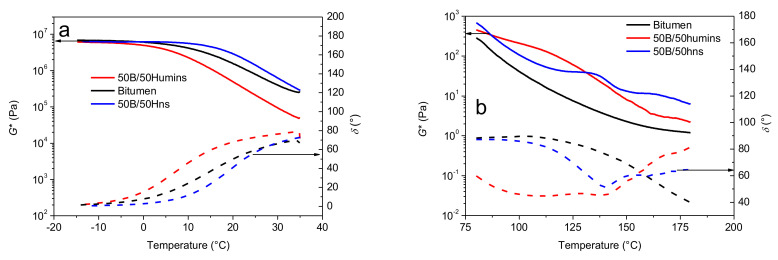
Variation of the complex modulus (solid line—left hand axis) and the phase angle (dashed line—right-hand axis) (**a**) between −15 and 35 °C and (**b**) between 80 to 180 °C for crude humins (black line), 50B/50Humins (red line), and 50B/50Hns (blue line).

**Table 1 polymers-14-01019-t001:** Assignments of the main FTIR peaks.

Wavenumbers (cm^−1^)	Assignment
3383	O-H stretching
2920	C-H out of phase stretching in methylene groups
2850	C-H in phase stretching in methylene groups
1725	C=O stretching in non-conjugated carbonyls
1670	C=O stretching in conjugated carbonyls
1605	Aromatic C=C stretching
1515	Furanic C=C stretching (with aldehyde)
1458	CH bending deformation in methylene
1375	CH_3_ symmetric deformation
1190	C-O stretching in ethers/esters
1020	C-O stretching in furan ring
805	C –H out of plane deformation in furan ring
760	C – H wagging in furan ring

## Data Availability

Not applicable.

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
