# Peer review of "Furanic Humins from Biorefinery as Biobased Binder for Bitumen"

_polymers, 2022, doi:10.3390/polym14051019_

Round 1

Reviewer 1 Report

This article shows no novelty. In addition, the authors use same materials and testing methods in previous paper published in 2019. I can not recommended it for publication.

Author Response

The reviewer 1 sent a single comment on this manuscript: ‘This article shows no novelty. In addition, the authors use same materials and testing methods in previous paper published in 2019. I can not recommended it for publication.

Starting from this assessment it is complicated to change or improve the paper.

We published two articles in 2019 related to humins

1/ A. Sangregorio, N. Guigo, E. de Jong, N. Sbirrazzuoli. Kinetics and Chemorheological Analysis of Cross-Linking Reactions in Humins. Polymers, 2019, 11 (11), 1804.

2/ A. Sangregorio, N. Guigo, J. C. Van der Waal, N. Sbirrazzuoli. All 'green' composites comprising flax fibres and humins' resins. Composites Science and Technology, 2019, 171, 70 – 77.

These two papers from 2019 have nothing in common with the newly submitted article as they deal with humins self-crosslinking (only humins were studied – see 1) or humins in combination with flax fibres (see article 2) to prepare composites. In the present article, we highlight the potential of using humins in combination with bitumen from crude-oil refining.  None of these articles from 2019 were dealing with humins/bitumen. To our knowledge this scientific article is the first one on humins/bitumen mixture - except one patent whose Ed de Jong (see ref. 24 van Klink, G.P.M.; de Jong, E. Asphalt composition comprising humins obtained from dehydration of carbohydrates. PCT patent application 2018, WO 2018/135941 Al.), one of our co-author, is inventor. However this patent does not contain any data which are shared in this publication. In addition, we present for the first time HnS which are the water non-soluble part of humins that were compared with raw humins for their capacity to mix with bitumen. It is the first time as well that HnS is employed in a scientific paper.

To conclude the novelty of this article cannot be discussed and we encourage the editorial board to simply verify our claims by checking the previous articles on humins (ours and others) compared to that one.

Reviewer 2 Report

Dear Authors,

The reviewed article is interesting. You undertook the analysis of an interesting research issue. You used a number of research methods, the results of which you interpreted correctly.

However, I would like you to respond to the following comments:

  1. the literature needs improvement. There are many invalid links in this article that cannot be identified.
    1. p.2, l.59; p.2 l.71; p.7 l. 234; p.8 l.275 and so on. Please check the entire article in this regard.
  2. Figures 1 and 2 require a more detailed description of the bands, please describe at least the positions of the most important bands. In my opinion, it is not enough to describe them in the text of the article.
  3. Figure 3 - This form adds nothing to the article. Please mark the effects that you describe in the text.
  4. There is no discussion of the magnitude of change in mass of the samples in the TGA analysis (Fig. 3). Such an analysis should be described in detail. Again, I believe that making a figure with only thermal curves is not enough to publish it in such an outstanding journal.
  5. The temperature ranges discussed in all thermal studies should be graphically marked. This definitely improves the quality of the article, but most of all, it will make it easier to read and analyze the results.
  6. When submitting thermal tests, it is usually also required to specify how the devices are calibrated. Please also complete this.
  7. Figures 6 and 7 show no measurement error. Please complete this as well.
  8. Figure 8 needs improvement. The arrows shown do not relate to specific results. Please think about a different, better way to represent the recorded changes in viscosity.
  9. research conclusions. I believe that the Summary has been prepared too broadly and needs to be improved. Especially in terms of listing all test results.

Best regards.

Author Response

  1/  the literature needs improvement. There are many invalid links in this article that cannot be identified.

        p.2, l.59; p.2 l.71; p.7 l. 234; p.8 l.275 and so on. Please check the entire article in this regard.

 Answer: Thanks for highlighting this. This issue comes from a word to pdf conversion. It is fixed in the revised version.

    2/ Figures 1 and 2 require a more detailed description of the bands, please describe at least the positions of the most important bands. In my opinion, it is not enough to describe them in the text of the article.

 Answer: We thanks the reviewer for his/her remark. We have noted the main peaks in Fig 1 and Figure 2. The assignments of these bands are detailed in Table 1 and it is thus easier to follow.

   3/ Figure 3 - This form adds nothing to the article. Please mark the effects that you describe in the text.

 Answer: Many thanks. To better guide the reader, we have highlighted the temperature region of the glass transition of aromatic waxes as well as the crystallization peak. The dash line represents the supposed baseline that help to better highlight the crystallization peak.

     4/ There is no discussion of the magnitude of change in mass of the samples in the TGA analysis (Fig. 3). Such an analysis should be described in detail. Again, I believe that making a figure with only thermal curves is not enough to publish it in such an outstanding journal.

Answer: We thank the reviewer for this remark. We have added discussion about the mass loss changes in the TGA and we have made more obvious the TGA curves by highlighting the different area

     5/ The temperature ranges discussed in all thermal studies should be graphically marked. This definitely improves the quality of the article, but most of all, it will make it easier to read and analyze the results.

Answer : As suggested by the reviewer we have marked in Figure 4 the different regions (i.e. LTO, FD, HTO) for bitumen degradation and the initial mass loss for 50B/HnS and 50B/humins samples (residual condensations). It should be easier for the reader to follow the discussion on Figure 4 data.

    6/ When submitting thermal tests, it is usually also required to specify how the devices are calibrated. Please also complete this.

 Answer: We have specified in the original version how the DSC was calibrated “Temperature, enthalpy and tau lag calibrations were performed using indium and zinc standards.” We have also detailed the calibration procedure for the TGA/SDTA as follow “

    7/ Figures 6 and 7 show no measurement error. Please complete this as well.

 Answer: We thank the reviewer for this suggestion. We have added the error bars in the Figure 5 and 6 (frequency sweep). The error bars correspond to the standard deviation from the means of 3 experiments. Note to the reviewer: as the scale is in log some error bar going down are higher than those going up; the size of some points are sometimes larger than the error bar themselves (due to the log scale somehow).

   8/ Figure 8 needs improvement. The arrows shown do not relate to specific results. Please think about a different, better way to represent the recorded changes in viscosity.

 Answer: There was a mistake in the Figure caption. These figures report the complex modulus and the phase angle (and not the viscosity). The arrow indicated the ordinate (left-hand axis for the modulus and right-hand axis for the phase angle). However it is true that these Figures were confusing. We have proposed another way to represent the modulus and phase angle variations.

    9/ research conclusions. I believe that the Summary has been prepared too broadly and needs to be improved. Especially in terms of listing all test results.

 Answer: We agree with the reviewer that the research conclusions were maybe too broad. We have re-written and expanded the conclusion to make a good overview of the different results and to connect that to future practical applications.

Reviewer 3 Report

This article is about furanic humins from biorefinery. This work is relevant and interesting, since biorefinery processes are promising and actively developing in the last decade. The work is written clearly and the data of physico-chemical analysis are well described, which adds to the quality of this work. I suggest the authors pay attention to the following points:
1. In all IR spectra, add the designations of the main functional groups.
2. Add the structural formulas of some of the humic substances that are expected to be in your process.
3. Throughout the article there is an expression "Error! Bookmark not defined.". This makes it difficult to evaluate the literature, as perhaps some of the references are missing.
4. Please increase the conclusions.
5. In some parts of the article, the following articles on this and related topics can be cited: 10.3390/polym13030372, 10.1021/acsapm.0c01390, 10.3390/polym12112732, 10.3390/polym13193267, 10.3390/polym12040750.
6. Unify the drawings, please.

In general, the work is of great interest and potential. I believe that it can be adopted after minor revision.

Author Response

This article is about furanic humins from biorefinery. This work is relevant and interesting, since biorefinery processes are promising and actively developing in the last decade. The work is written clearly and the data of physico-chemical analysis are well described, which adds to the quality of this work. I suggest the authors pay attention to the following points:

  1. In all IR spectra, add the designations of the main functional groups.

Answer: We thanks the reviewer for his/her remark. We have noted the main peaks in Fig 1 and Figure 2. We have added Table 1 with the assignments of the main functional groups.

  1. Add the structural formulas of some of the humic substances that are expected to be in your process.

Answer: Thanks for the suggestion. We have added in supporting information one Figure that represent the expected structure of furanic humins.

  1. Throughout the article there is an expression "Error! Bookmark not defined.". This makes it difficult to evaluate the literature, as perhaps some of the references are missing.

Answer: Thanks for highlighting this. This issue comes from a word to pdf conversion. It is fixed in the revised version.

  1. Please increase the conclusions.

 Answer : In the initial version, the research conclusions were maybe too broad. We have re-written and expanded the conclusion to make a good overview of the different results and to connect that to future practical applications.

  1. In some parts of the article, the following articles on this and related topics can be cited: 10.3390/polym13030372, 10.1021/acsapm.0c01390, 10.3390/polym12112732, 10.3390/polym13193267, 10.3390/polym12040750.

 Answer : We thank the reviewer for these suggestions and we have added most of these suggested papers in the manuscript.

 Unify the drawings, please.

 Answer : We have harmonized and unified the drawings within the manuscript.

Round 2

Reviewer 1 Report

Accepted in the current version.

Reviewer 3 Report

The authors managed to improve the work.

I believe that this work can be accepted for publication.